# Transcranial Pulsed-Current Stimulation versus Transcranial Direct Current Stimulation in Patients with Disorders of Consciousness: A Pilot, Sham-Controlled Cross-Over Double-Blind Study

**DOI:** 10.3390/brainsci12040429

**Published:** 2022-03-24

**Authors:** Alice Barra, Martin Rosenfelder, Sepehr Mortaheb, Manon Carrière, Geraldine Martens, Yelena G. Bodien, Leon Morales-Quezada, Andreas Bender, Steven Laureys, Aurore Thibaut, Felipe Fregni

**Affiliations:** 1Coma Science Group, GIGA Consciousness-GIGA Research, University of Liège, 4000 Liège, Belgium; a.barra@uliege.be (A.B.); mortaheb.sepehr@gmail.com (S.M.); manon.crre@gmail.com (M.C.); geraldine.martens@chuliege.be (G.M.); stevenlaureys01@gmail.com (S.L.); 2Centre du Cerveau, University Hospital of Liège, 4000 Liège, Belgium; 3Department of Physical Medicine and Rehabilitation, Spaulding Rehabilitation Hospital, Harvard Medical School, Boston, MA 02114, USA; ybodien@mgh.harvard.edu (Y.G.B.); jmorales-quezada@mgh.harvard.edu (L.M.-Q.); 4Department of Neurology, Therapiezentrum Burgau, Kapuzinerstrasse 34, 89331 Burgau, Germany; m.rosenfelder@therapiezentrum-burgau.de (M.R.); andreas.bender@med.uni-muenchen.de (A.B.); 5Clinical and Biological Psychology, Institute of Psychology and Education, Ulm University, 89081 Ulm, Germany; 6Physiology of Cognition Lab, GIGA-Consciousness, University of Liège, 4000 Liège, Belgium; 7Center for Neurotechnology and Neurorecovery, Department of Neurology, Massachusetts General Hospital, Harvard Medical School, Boston, MA 02114, USA; 8Neuromodulation Center, Spaulding Rehabilitation Hospital, Harvard Medical School, Boston, MA 02114, USA; felipe.fregni@ppcr.org; 9Department of Neurology, Ludwig-Maximilians University (LMU), 81377 Munich, Germany; 10Joint International Research Unit on Consciousness, CERVO Brain Research Centre CIUSS, University Laval, Quebec, QC G1E1T2, Canada

**Keywords:** minimally conscious state, coma, non-invasive brain stimulation, electrophysiology, neuromodulation, randomized controlled trial

## Abstract

Transcranial direct-current stimulation (tDCS) over the prefrontal cortex can improve signs of consciousness in patients in a minimally conscious state. Transcranial pulsed-current stimulation (tPCS) over the mastoids can modulate brain activity and connectivity in healthy controls. This study investigated the feasibility of tPCS as a therapeutic tool in patients with disorders of consciousness (DoC) and compared its neurophysiological and behavioral effects with prefrontal tDCS. This pilot study was a randomized, double-blind sham-controlled clinical trial with three sessions: bi-mastoid tPCS, prefrontal tDCS, and sham. Electroencephalography (EEG) and behavioral assessments were collected before and after each stimulation session. Post minus pre differences were compared using Kruskal–Wallis and Wilcoxon signed-rank tests. Twelve patients with DoC were included in the study (eight females, four traumatic brain injury, 50.3 ± 14 y.o., 8.8 ± 10.5 months post-injury). We did not observe any side-effects following tPCS, nor tDCS, and confirmed their feasibility and safety. We did not find a significant effect of the stimulation on EEG nor behavioral outcomes for tPCS. However, consistent with prior findings, our exploratory analyses suggest that tDCS induces behavioral improvements and an increase in theta frontal functional connectivity.

## 1. Introduction

Patients surviving brain injury may progress through several pathological states before eventually recovering consciousness. These states are commonly referred to as disorders of consciousness (DoC). DoC are comprised of a spectrum of states showing impaired arousal and awareness at variable levels. They include coma, the unresponsive wakefulness syndrome/vegetative state (UWS/VS), and the minimally conscious state (MCS). The UWS involves the recovery of eye-opening but no evidence of awareness of self or the environment [1,2]. When the patient recovers reproducible behavioral signs of awareness of self-and/or the environment, he or she is no longer said to be unconscious but in a MCS [3]. Finally, if the patient recovers functional communication and/or functional use of objects, he or she is said to have emerged from the MCS (eMCS) [3].

The non-communicative nature of DoC patients makes the research on diagnosis and treatment of these patients as important as it is challenging. To this day, even though significant progress has been made in understanding the neural correlates of DoC, the available treatments for DoC patients remain limited [4].

In addition to pharmacologic interventions such as Amantadine [5,6], the scientific community has witnessed the development of neuromodulation treatment approaches. Neuromodulation is a broad term that refers to different brain stimulation techniques that can be either invasive or non-invasive. It is now used to treat several neuropsychological conditions [7,8,9] as an alternative to, or for people that are resistant to pharmacological treatments. Some neuromodulation approaches have proven to be such valid options that they are currently FDA approved in several countries [10,11,12]. Non-invasive brain stimulation such as transcranial current stimulation (tCS) is a branch of neuromodulation of particular interest for DoC as it is safe, inexpensive, easily integrated into rehabilitation and hospital environments, and does not require the patient’s active participation. More specifically, transcranial direct-current stimulation (tDCS) uses a weak constant electrical current sent via, at least, one anode and one cathode. Anodal stimulation is thought to increase neuronal excitability by facilitating the action potential release and modification of the excitability of NMDA receptors [13,14]. Transcranial pulsed current stimulation (tPCS) is another type of tCS that uses a unidirectional pulsed flow of current [15]—as opposed to the continuous one of tDCS—and has been hypothesized to induce effects not only by modulating the polarization of the membranes, but also by some carrying proteins and neurotransmitters (e.g., catecholamines, 17-ketosteroids and endorphins release) [16].

TDCS has previously showed to produce transient improvements in the behavioral responsiveness of DoC patients. Notably, the first sham-controlled double-blind, randomized crossover study explored the effects of a single session of prefrontal tDCS in patients with DoC [17]. Subsequent tDCS studies targeting the prefrontal cortex reported similar results [18,19,20,21]. More recently, tDCS has been tested with different targets such as precuneus [22], the motor cortex [23], the posterior parietal cortex [24], and the frontoparietal network [25], but so far, the brain region that has shown more consistently positive results seems to be left prefrontal cortex [26]. Effects of tDCS have been most frequently observed in patients in MCS rather than UWS [17,22,27]. Neurophysiological studies have also investigated the effect of tDCS, mainly using EEG. TDCS over the left prefrontal cortex was reported to increase theta and alpha power (as measured by EEG) in several studies, and to decrease delta power [27,28]. Furthermore, tDCS may improve EEG background organization (i.e., reaching a normal background activity) [18] and stronger P300 event-related potentials [22].

TPCS has not yet been studied as a treatment for patients with DoC and only a few studies have explored it in patients with neurological conditions (e.g., Parkinson’s disease) [29,30]. However, tPCS can enhance speech comprehension in healthy subjects [31] as well as facilitate arithmetic tasks [32] and enhance motor skills and cognitive functions [33], suggesting that this stimulation may also promote recovery in patients in MCS. Furthermore, it was shown, using computational modeling, that TPCS can modulate subcortical neural circuits [33,34]. Previous modeling studies have found that tPCS can modify the electrical activity of cortical and subcortical structures [34], and improve frontal [35] and interhemispheric neuronal connectivity [36]. For this reason, it has been hypothesized that tPCS could reach deeper brain structures compared to tDCS.

As this was a pilot study using both tPCS and tDCS in patients in DoC, our first objective was to evaluate the feasibility of a double-blind randomized controlled trial using these two different stimulation techniques, and assess the safety of tPCS in this population. In addition, we aimed to compare the neurophysiological and behavioral effects of a single session of tDCS and tPCS against the sham. To do so, we first evaluated the change in neurophysiological parameters (i.e., quantitative EEG (qEEG) power and functional connectivity), especially in the theta and alpha bands over the frontal and parietal areas. Based on the literature reported above, we hypothesize the following differences between the stimulation conditions: tPCS will enhance cortical activity in the frontal and parietal areas within the theta and alpha bands compared to the sham; tDCS will enhance cortical activity in the frontal areas within the theta band compared to the sham. Finally, we compared the Coma Recovery Scale-Revised (CRS-R) total scores and CRS-R modified index between conditions. We hypothesized that both tPCS and tDCS would promote recovery of signs of consciousness on the CRS-R compared to the sham.

## 2. Materials and Methods

This was a pilot double-blind sham-controlled randomized crossover clinical trial exploring the effects of two different stimulation techniques (tDCS and tPCS) against the sham stimulation on behavioral and neurophysiological outcomes in patients with at least one sign of consciousness (i.e., CRS-R diagnosis of MCS or eMCS).

Study design: Randomized placebo-controlled, multi-center double-blind crossover pilot study.

Participants: Written informed consent in accordance with the Declaration of Helsinki was obtained by the legal representative of each patient. The ethics committee approved this study of the University Hospital of Liège, and by each local ethic committee as needed (Clinicaltrial.gov: NCT03115021). All patients were included while in rehabilitation centers (Wallonia, Belgium); the amount of rehabilitation they received remained unchanged during the study protocol. Participants were pre-screened by their physicians and, if they met the inclusion criteria, they were assessed for eligibility. 

The inclusion criteria were the following: (a) age between 18 and 75; (b) history of acquired traumatic or non-traumatic severe brain injury leading to a DoC; (c) a minimum of two CRS-R assessments, performed by a trained clinician or research staff before inclusion, detecting at least one, but not all, signs of consciousness (MCS/eMCS); and (d) in stable medical condition (e.g., no infection, intubation, recent hospitalization, escalating or de-escalating medications). 

Participants were excluded in the following cases: (a) history of major neurologic or psychiatric comorbidity present at the time of enrollment; (b) evidence or report of untreated seizure; (c) presence of a metallic implant or implanted electronic brain medical devices (e.g., pacemaker); and (d) history of cranioplasty without healed cranial flap, based on medical records. Note that the reason why we excluded patients with untreated epilepsy is that previous tES trials on patients with DoC have enrolled patients treated for epilepsy and have not found any side effects (e.g., [17,20]).

Blinding: Patients received all three stimulation sessions (tPCS, tDCS, sham) in random order in a 1:1:1 ratio. The patient, the researcher administering the treatment, and the researcher evaluating the response to the treatment were all blinded to the condition. A researcher not involved in data collection created sealed randomization envelopes and assigned them to each patient in a chronological order. To ensure blinding of both the researchers and the patient, both systems (i.e., electrodes for tPCS and for tDCS) were mounted for each session; one researcher launched the stimulation (according to the coded order in the sealed envelope) while the researcher involved in behavioral assessment remained away from the patient’s room to ensure full blinding. Moreover, the EEG analysis was performed with a blinding code: conditions (i.e., tPCS, tDCS and sham) were renamed with a numerical code that was revealed to the researcher only when the analysis was finished.

Study protocol: Each patient was evaluated by a trained professional at least twice with the CRS-R before being enrolled in the study to ensure that the patient met the diagnostic inclusion criteria. The protocol included three sessions and each patient completed all three sessions in a random order: (i) tPCS, (ii) tDCS, and (iii) sham. Each session was separated by a minimum of five days to avoid any carryover effect. This washout period was established based on similar studies and on the fact that the effects of a single session of tDCS should not last more than a few hours after the end of stimulation [37]. Each session was structured as follows: the CRS-R was performed, and 10 min of resting EEG were recorded. Twenty minutes of stimulation were launched. After the stimulation, the CRS-R and the EEG were collected again. For a schematic view of the protocol, see Figure 1.

Interventions: For prefrontal tDCS, we used a NeuroConn DC-Stimulator PLUS device (Ilmenau, Germany) delivering a maximum of 2 mA of current for 20 min. The anode was placed over F3 (International 10/20 EEG System), corresponding approximately to the left prefrontal cortex. The cathode was placed over the contralateral orbitofrontal area. Both anode and cathode were round rubber electrodes with a diameter of 4 cm (12 cm^2^). For bi-mastoid tPCS, we used the NeuroConn DC Stimulator MOBILE (Ilmenau, Germany), which delivered a random frequency between 6–10 Hz with a biphasic current of 2 mA peak to peak for 20 min. The rubber electrodes were placed over the mastoids. These squared electrodes covered an area of 16 cm^2^ (4 × 4 cm). Both tPCS and tDCS electrodes were fixed using a conducting paste (Ten20, Weaver and Company, Aurora, CO, USA) to reduce impedances and were placed under the EEG cap. For sham, the same parameters were set as for the active condition (both tDCS and tPCS montages), but the device automatically turned off after 30 s to simulate the initial tingling sensation of the active current.

Assessments: As this was a pilot study, our primary goal was to evaluate the feasibility of a randomized double-blind clinical trial by comparing two different types of tCS. We defined feasibility as successfully carrying out the pilot study on the sample (i.e., successful mounting of the systems, double blinding, execution of the sessions, no adverse events during the stimulations). We assessed the side-effects of the stimulation by observing the patients before, during, and after each session of stimulation (both sham and active) with a particular attention to redness of the skin, irritation of skin, seizures, facial and verbal signs of pain and/or discomfort, which were reported on the patient’s case report form. Caregivers and families were asked to report to the researchers if they noticed any side-effect in the days following the stimulation. Moreover, to further investigate the safety of the stimulations, a trained professional checked the EEG recordings post-stimulation for ictal and irritative abnormal activities.

Our secondary goal was to evaluate the effects of bi-mastoid tPCS and left prefrontal tDCS on neurophysiology (EEG) and behavioral measures (CRS-R). To record EEG, we used a portable device developed by NeuroConn (Ilmenau, Germany). The NeuroPrax-Tes 2.6.17 system is a full-band DC-EEG system that is compatible with tCS stimulation and allows for the recording of EEG at high sampling rates. This system is composed of a cap with 27 gel-based (25 recording electrodes and two references) electrodes, one amplifier, and one portable computer. The EEG was mounted at the beginning of the session for every patient by the researchers and the stimulation electrodes were placed under the EEG cap. The recording lasted 10 min, and the sampling rate of the recordings was kept as the default at 4000 Hz. The reference electrodes (one reference and one ground) were placed behind the patients’ ears with adhesive electrodes. For the behavioral assessment, we used the CRS-R, which is composed of six subscales organized in a hierarchical order where lower scores represent lower behavioral capabilities [38]. A trained researcher performed the CRS-R, before and after each stimulation, in the patient’s room, which was kept quiet, and the patient was made as comfortable as possible, either in bed or in a wheelchair.

### 2.1. Data Analyses

Analysis of the crossover effect: First, we performed a Kruskal–Wallis test on the baselines (i.e., pre-stimulation) to compare sham, tPCS, and tDCS and make sure that they did not differ at the baseline. We performed this analysis on both the EEG data and behavioral data. 

EEG preprocessing: EEG data were preprocessed using custom scripts in MNE-Python [39]. Data were filtered with a high-pass filter and low-pass filter set at 1 Hz and 30 Hz, respectively. The sampling rate was downsized from 4000 to 500 Hz. The signal was further divided into 2 s epochs. Visual inspection was used to identify and remove artifactual signal epochs and noisy channels. An independent component analysis (ICA) based on the Infomax algorithm [40] was used to remove eyeblink and muscle artifacts (based on their temporal, frequency and spatial distribution). Afterwards, bad channels were interpolated using spherical interpolation, and data were re-referenced to the average of all electrodes (i.e., average referencing). If the extracted clean epochs were not sufficient (i.e., <90 epochs of 2 s) to perform the statistical analyses, the patient was discarded from the EEG analyses. One patient was discarded from EEG analyses for this reason (see Figure 2 for details).

EEG power: Power spectral density (PSD) was calculated using custom scripts in MNE-Python [39] for each band (i.e., delta 1–4 Hz, theta 4–8 Hz, alpha 8–13 Hz, and beta 12–30 Hz), for each electrode, and averaged over the epochs with a trimmed mean that allowed us to exclude outliers. Welch’s method was used to extract the resulting averaged power bands with a 1 s window overlapped at 50% and a frequency resolution of ½ Hz. We focused the analyses on frontal electrodes (i.e., F3, F4, F7, F8, Fc1, Fc2, Fc5, Fc6, Fp1, Fp2, Fz) and parietal electrodes (i.e., Cp5, Cp6, P3, P4, Pz) as this follows the mesocircuit hypothesis on the recovery from DoC [41,42].

EEG connectivity: The connectivity between pairs of channels was calculated for each band, in channels encompassing the frontal and parietal region, using the debiased weighted phase lag index (dwPLI). First, data were analyzed in the time-frequency domain regarding power and current spectral density, using the spectral transfer function. Then, the dwPLI was computed from the peak frequency in any pair of electrodes. For an in-depth description of the analyses, see [43]. The resulting connectivity matrix was collapsed to each frequency band (i.e., delta, theta, alpha, beta) and the median voltage was calculated for each frequency band. We focused the analyses on the frontal and parietal regions using the same electrodes as for power, based on the literature regarding the mesocircuit model [41,42].

### 2.2. Statistical Analyses

The feasibility of the use of tPCS in a randomized double-blind clinical trial was inferred from the success of the present pilot study. The safety of the stimulation techniques—with a focus on tPCS, as the safety of tDCS has already been demonstrated [17]—was monitored by reporting any side-effects observed by the researchers, caregivers, or family before and after each session and by screening post-stimulation EEG recordings for the presence of abnormal (ictal/irritative) activity.

EEG data were analyzed using RStudio (version 1.2.5001; RStudio, Inc., Boston, MA, USA). We performed Kruskal–Wallis analyses on the pre and post differences (i.e., delta—Δ) of the three conditions. The power differences pre and post for each electrode were averaged for each patient to account for the subject’s variability. For instance, when performing the analysis for frontal regions, we took the power values for each frontal electrode (i.e., FP1, F3, F7, Fc1, Fc5, FP2, F4, F8, Fc6, Fc2, FZ) post-stimulation, subtracted them from the pre-stimulation values, and then calculated the average of those differences for each patient. We extracted the power data for each power band, but we focused the statistical analyses on alpha and theta over the frontal and parietal regions, based on our a priori hypothesis. The same analysis was performed for connectivity data. *p* values were considered significant after correcting for multiple comparisons for the three conditions, three regions, and two power bands using a Bonferroni correction (*p* < 0.003). Behavioral data was analyzed using RStudio. We performed a Kruskal–Wallis test to assess the main effect of the intervention on the three conditions’ pre- and post-differences (delta) using CRS-R total scores. We also transformed CRS-R total scores in the modified index using the method described by Annen and colleagues [44], since this index is more sensitive to changes and directly linked to diagnosis (with a cutoff at 8.315 that distinguishes patients in UWS from patients in MCS). With this method, the CRS-R modified index is calculated by combining scores for reflexes and cognitive behaviors of every CRS-R subscale (only the highest observed behavior for each subscale) and computes the score from a transposition matrix. We then performed the same analyses on modified index deltas. We conducted post-hoc tests to determine whether the main effect of treatment was significant with the Wilcoxon signed-rank test. *p* values were considered significant after correcting for multiple comparisons for the three conditions using a Bonferroni correction (*p* < 0.0125).

Further exploratory analysis: Given the results we obtained on both behavioral and EEG data, we performed a posteriori exploratory analysis investigating whether tPCS was similar to the sham and different from tDCS. We therefore compared the sham and tPCS combined [tPCS + sham] to tDCS. This was based on the similarities we observed between tPCS and the sham and we performed this analysis on RStudio for both the behavioral data (CRS-R total scores and CRS-R modified index) and the EEG data (qEEG and connectivity of alpha and theta, frontal and parietal) with an unpaired Wilcoxon signed-rank test.

## 3. Results

We enrolled 13 patients, but one patient died right after enrollment; therefore, 12 patients were included in the analyses (10 MCS, 2 EMCS, 8 F, 4 TBI, 50.3 ± 14 y.o., 8.8 ± 10.5 months post-injury). Eleven of these patients received all three sessions, whereas one received only tPCS and sham stimulations due to a decompressive craniectomy without flap replacement under the stimulation site that took place briefly after enrolling the patient. Details on the patients’ enrollment can be found in the flow chart in Figure 1. Demographic details of the patients can be found in Table 1.

Feasibility and Safety: The feasibility of performing a randomized clinical trial in a double-blind design, comparing two stimulations to the sham, was demonstrated with this pilot, in which patients completed the assigned sessions without any side-effects. Blinding was achieved by mounting both devices every time, regardless of the condition. Safety of tPCS in patients with DoC was demonstrated as we did not observe any side-effects in any of the patients during the stimulation sessions. Moreover, when EEG recordings post stimulation were inspected, no abnormal activity was detected. At the individual level, for one subject (i.e., ID01), the family reported trouble falling asleep and longer wakefulness for one night after receiving active tPCS. No other change after stimulation was reported for any of the subjects.

Crossover effect: We did not find any significant difference between groups at the baselines before each session for EEG nor for the behavioral data.

EEG: The Kruskal–Wallis test showed no significant difference between the stimulation conditions for EEG power for the alpha and theta bands (see Annexes for details). Frontal and parietal theta band connectivity did not show any significant differences. See Table 2 and Figure 3a,b for details. 

Behavioral: For the behavioral data, we did not find any significant changes in CRS-R scores at the group level for any of the conditions. This analysis was performed with the Kruskal–Wallis test on both the CRS-R total score (Chi squared= 1.47, df = 2, *p*-value = 0.47) and on the modified index (Chi-squared= 4.11, df = 2, *p*-value = 0.12). See Table 2 and Figure 3c,d for details.

Exploratory analysis: When comparing each technique separately with the Wilcoxon signed-rank test, we found that for both the CRS-R total score and for the modified index tPCS and sham were very similar, almost reaching *p*-values of 1. Therefore, to increase statistical power and better investigate the differences between tPCS and tDCS, we conducted an exploratory analysis comparing [tPCS + sham] to tDCS with an unpaired Wilcoxon signed-rank test. We found a significant difference between tDCS vs. [tPCS + sham] conditions for theta connectivity in the frontal regions (W = 59, *p* = 0.04) (for individual results, see Table 3; for visual schematization of theta frontal data, see Figure 4a).

## 4. Discussion

To our knowledge, this is the first clinical trial on the use of tPCS stimulation in patients with DoC. Altogether, this work demonstrated the safety of tPCS in patients with DoC and the feasibility of combining two tCS techniques in randomized controlled double-blind clinical trial. 

The successful completion of this pilot study, without major issues nor side-effects to the stimulations, opens the door for future studies to investigate the use of tPCS as a therapeutic intervention for patients with DoC. Based on our outcome measures, we did not find any significant main effect of one session of bi-mastoid tPCS compared to left prefrontal tDCS or to sham on the behavioral or electrophysiological outcomes.

When comparing all three types of stimulation, we noticed that tPCS seemed to be similar to the sham (with *p* values close to 1, similar medians and IQRs). Therefore, in an exploratory analysis, we merged the two groups [tPCS + sham] and compared them to tDCS to increase the statistical power. Doing so, we observed an improvement following tDCS at the neurophysiological and behavioral levels compared to [tPCS + sham]. These analyses were exploratory and must be interpreted with caution.

According to our functional connectivity analyses, there was no significant effect of bi-mastoid tPCS stimulation on theta and alpha bands, as predicted. Alpha functional connectivity in the frontal and central regions has been reported to be boosted following tPCS in a study on healthy controls using the same frequency as in this study [45]. Although we used the same stimulation bandwidth (6–10 Hz), we did not observe a significant increase in frontal alpha functional connectivity compared to the other stimulation groups. It is important to note that Morales-Quezada and colleagues obtained these results on healthy participants, while it is well known that patients with prolonged DoC likely suffer from altered oscillatory patterns which could have limited the effect of 6–10 Hz tPCS. On the other hand, tPCS applied with a frequency between 1–5 Hz was shown to modulate theta power and fronto-temporal connectivity in healthy individuals [36]. Therefore, to increase theta band power in DoC patients, targeting this frequency range (1–5 Hz) could be more effective [46]. Another parameter that could explain the absence of effects is that the number of sessions of tPCS was too small to induce any significant behavioral and neurophysiological changes. This may also apply to tDCS given its relatively small effect size [47].

Regarding tDCS, results from our exploratory EEG analyses tend to confirm recent studies. We found an increase after left prefrontal tDCS for theta frontal connectivity. This result not only replicates those of previously mentioned studies [27,28], but also supports the idea that theta band connectivity in EEG might serve as a biomarker of responders to non-invasive brain stimulation with tDCS, as proposed in previous studies [4,48]. Thibaut and colleagues hypothesized that prefrontal tDCS could be a powerful tool to stimulate these under-activated theta-band network connections, resulting in clinical improvements at the same time. At the group level, this assumption is consistent with our results. The tDCS treatment was associated with an increase in frontal theta connectivity and an improved behavioral responsiveness (i.e., CRS-R modified index) compared to the [tPCS + sham] treatment. Testing for the presence at the baseline of theta band connectivity could also be a good predictor of the patient’s likelihood of benefitting from the stimulation. For this reason, it might be worth adding this test to the inclusion criteria for future studies, to better target patients who might respond to tCS.

In the current literature, most studies have also generally found an effect of tDCS on alpha band connectivity [27,28,48] and we could not reproduce this result in the present study. It should be noted that Hermann and colleagues did not find effects on isolated alpha connectivity but on cross-frequency theta-alpha functional connectivity (e.g., 4–10 Hz), which was pronounced in responders parieto-occipitally when compared to non-responders. Thus, investigating combined frequency bands and intra- as well as inter- hemispheric functional connectivity seems promising for further tDCS/tPCS trials in DoC patients.

Regarding the behavioral effects, we did not find an effect of tPCS stimulation on the CRS-R total score and modified index. Although we know from healthy subject studies that tPCS can increase performance in attention-switching tasks [49] we did not observe similar effects in DoC patients. Regarding arousal, in a study targeting patients with chronic visceral pain, tPCS or combined tDCS-tPCS stimulation did not influence self-reported sleepiness [50]. Similarly, wakefulness was not significantly changed by the stimulation in our study. Arousal was reported to be higher in one patient (caregivers reports), however, this was only observed at the individual level.

The absence of behavioral effects could be explained by several factors. The first one is the small sample size, which might have led to underpowered statistical tests. As we did not use a neuronavigation system, electrode location might have been imprecise. In addition, patients received a single session of tCS, when we know that repeated sessions are needed to induce significant and long-lasting behavioral effects. Finally, the overall length of each session, which duration was about 3 h, might have caused an increase in the patients’ fatigue. Further speculations regarding the effectiveness of tPCS in DoC patients cannot be made yet, as this is, to our knowledge, the first study of tPCS application in this field. 

In the exploratory analysis of this study, we found that tDCS slightly improved the performance of patients on the CRS-R modified index compared to the [tPCS + sham]. We addressed this finding to the fact that there is already evidence for the efficacy of left prefrontal tDCS as a tool to enhance the responsiveness of patients with DoC [28]. Other studies showed positive behavioral effects of tDCS when stimulation was applied for at least one week: Estraneo and colleagues found clinical and electrophysiological improvements in five out of 13 patients; however, these changes lasted over several months [18]. Another study also applying five days of prefrontal tDCS found a significant effect of active compared to sham treatment even one week after treatment, which also produced long-lasting clinical improvements, as measured with the CRS-R [21]. Finally, tDCS treatment over four weeks applied at home had a significant effect on 27 patients with DoC, demonstrating the stability of its effect even if applied in non-clinical settings [20]. 

### Study Limitations

The biggest limitation of our study was the sample size of twelve patients, which decreased our statistical power to detect a treatment effect. The reason behind this choice of sample size was that this combination of stimulation montages is technically challenging for researchers and very demanding for patients, as the protocol lasts about 3 h including EEG/tCS placement and two CRS-R assessments. For this reason, this was planned to be a pilot study to test safety and feasibility.

Regarding the lack of neuronavigation, it has been suggested [25] that montages tailored to the patients’ specific brain lesions might have more probabilities of success, since the structural preservation of brain areas targeted by the stimulation seem to be a crucial factor to respond to stimulation [51]. In the present study, we could not use this approach as not all patients underwent recent neuroimaging investigation that could have helped us to adapt tCS montages to each patient. However, this should definitely be considered for future studies.

Regarding the stimulation parameters, future studies should consider optimizing them (e.g., stimulation target, stimulation parameters, number of sessions) for patients with DoC. In our case, for instance, although the frequency that was chosen for tPCS was based on modeling studies, it might not have been optimal for DoC patients. Based on the neural noise hypothesis, spontaneous frequencies need to be present in the brain at the baseline in order to be entrained by neuromodulation [52]. A frequency-range between 6 and 10 Hz (as the one used in our study) was able to produce EEG changes in healthy participants [53]. We can thus hypothesize that stimulation within this range in patients with DoC, which demonstrate predominant activities within the delta-theta range, failed to produce neurophysiological results. In this context, targeting lower frequencies (e.g., 1–5 Hz) could be more effective for patients with DoC. In addition, the placement of the electrodes might have influenced the level of cortical activation [34]. Future studies might look at different modalities of stimulation targeting, for instance, the prefrontal cortex instead of the mastoids. 

Regarding the EEG functional connectivity results, another drawback to mention is that the Mohawk method pipeline [43] for the extraction of functional connectivity values has not yet been validated with a 27 electrodes system and was initially developed for a 265 channel high-density EEG. For this study, the pipeline has been adapted according to the Neuroconn EEG system, so there could have been unpredicted weaknesses in the pipeline derived from the number of electrodes. However, Chennu and colleagues reported that the median dwPLI connectivity was a relatively stable index against reducing the number of electrodes down to at least 11 electrodes. Another study on multichannel tDCS to reduce hypertonia in DoC patients found an effect of the treatment on beta connectivity with only eight EEG recording channels [54].

Another limit to our study is probably the choice of applying only one session of stimulation and not repeated sessions. As above-mentioned, repeated session studies have shown stronger and longer behavioral effects. However, they take longer to complete and the dropout rates are higher than those of single session studies. As this was planned to be a feasibility study, we decided to start with a single stimulation session. However, future studies should explore the effects of repeated sessions of tPCS.

Finally, fluctuating arousal and fatigue could have induced a bias during the behavioral and electrophysiological measurements. Each session lasted around 3 h, and for patients with DoC, whether in bed or in a wheelchair, such a long protocol is likely to induce fatigue. This, in turn, can impact both neurophysiological and behavioral outcomes. Maintaining arousal is a key element to attention and awareness [55], which is known to be impaired in DoC patients [56]. The behavioral assessments were always preceded by an arousal protocol, as specified in the guidelines of the CRS-R [38]. Nevertheless, in the future, short protocols focusing on behavioral or EEG outcomes should be preferred to avoid fatigue.

## 5. Conclusions

This study showed, for the first time, that tPCS is a feasible and safe technique for the treatment of patients with DoC. Although we did not find a significant effect of tPCS compared to sham on the level of consciousness or on electrophysiological outcomes, our findings provide a foundation for future studies to continue investigating the efficacy of tDCS and tPCS stimulation to promote the recovery of consciousness after a severe brain injury.

## Figures and Tables

**Figure 1 brainsci-12-00429-f001:**
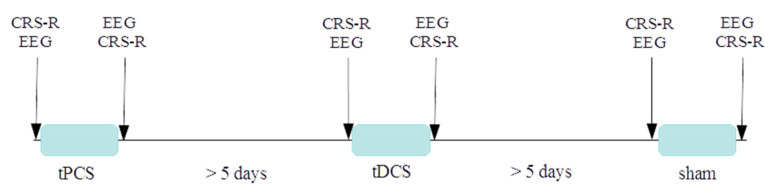
Schematic representation of the protocol. Note that the order of the sessions was randomized for each patient (the order of sessions for each patient can be found in Table 1). During each session both devices were mounted together with the EEG cap, to keep the same montage across sessions and allow blinding.

**Figure 2 brainsci-12-00429-f002:**
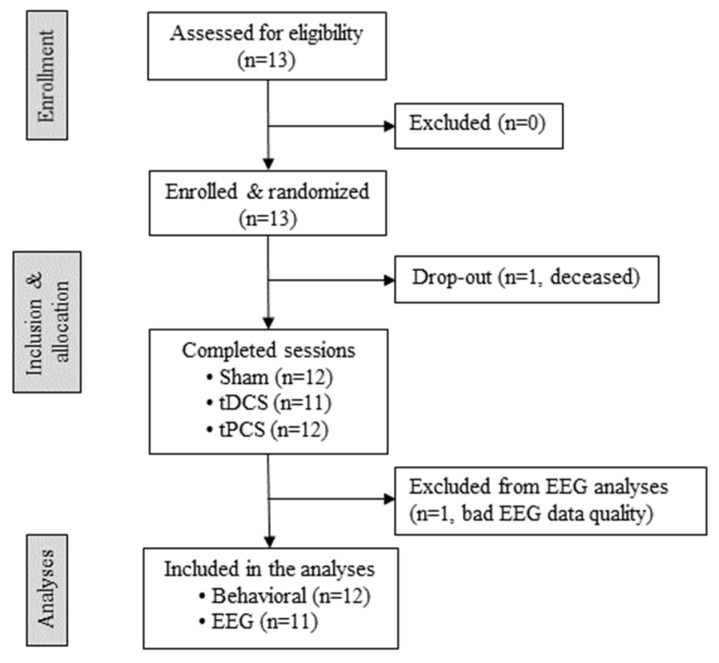
Flow chart of patient enrollment. The specifics on patients’ enrollment, allocation, and analysis are reported. We were in contact with the collaborating centers, and when a patient matched the inclusion criteria, we were contacted by their health care team to go screen the patient and confirm eligibility. This is why this could be considered as a convenience sample (i.e., non-probability sampling, typical of pilot studies). Thirteen patients were screened for eligibility. All patients were randomly allocated to a specific order of the three sessions (see Table 1 for details). One patient died after the randomization for medical reasons external to the study. Eleven patients received all three sessions, one patient received tPCS and sham, but no tDCS due to decompressive craniectomy that took place after the first two sessions. One patient was not included in the EEG analysis due to the poor quality of the EEG data. Transcranial direct current stimulation (tDCS), transcranial pulsed current stimulation (tPCS), electroencephalography (EEG).

**Figure 3 brainsci-12-00429-f003:**
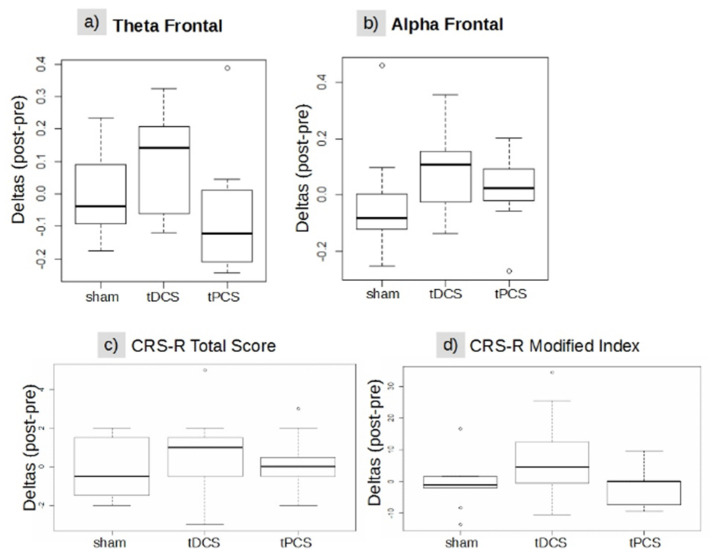
(**a**–**d**) Boxplot group differences between treatment conditions. (**a**) Functional connectivity in the frontal theta band, (**b**) in the frontal alpha band, (**c**) CRS-R total score, and (**d**) CRS-R modified index. The rectangles represent the interquartile range 25 and 75; the bold horizontal lines inside the rectangles are the medians of each group, empty circles indicate potential outliers (minimum or maximum value in the data) whereas the whiskers are calculated by the default boxplot R function and represent 1.5 distance from IQR 25 and IQR 75. CRS-R = Coma Recovery Scale-Revised, Sham = sham treatment, tDCS = transcranial direct current stimulation treatment, tPCS = transcranial pulsed current stimulation treatment, IQR = interquartile range.

**Figure 4 brainsci-12-00429-f004:**
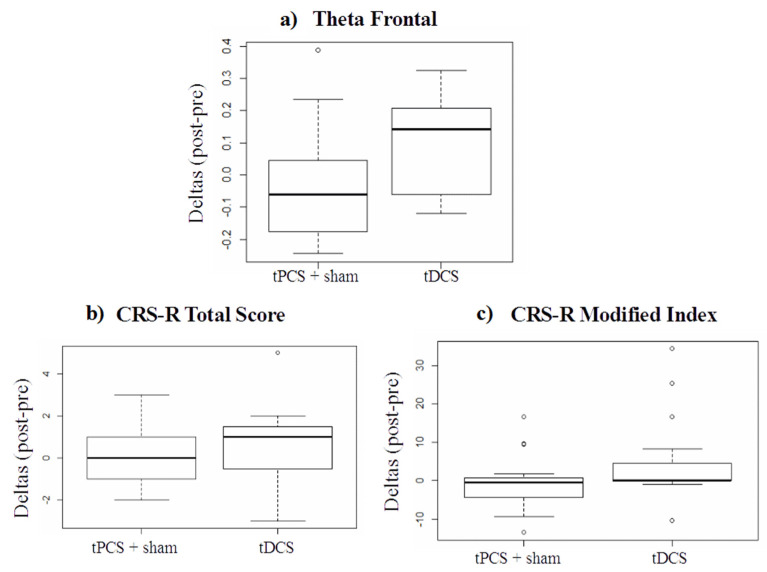
Group differences between treatment conditions on ([tPCS + sham]) and tDCS: (**a**) on theta frontal functional connectivity, (**b**) on CRS-R total score, and (**c**) on CRS-R modified index. Calculated with unpaired Wilcoxon signed-rank test. tPCS = transcranial pulsed current stimulation, sham = sham, tDCS = transcranial direct current stimulation.

**Table 1 brainsci-12-00429-t001:** Clinical characteristics of the sample and behavioral results. Age is reported in years. Diagnosis refers to the two screening CRS-R before enrollment (minimally conscious state—MCS, emergence from minimally conscious state—eMCS. Etiology reported (traumatic brain injury—TBI). Time since injury (TSI) reported in months. Total CRS-R scores pre- and post-stimulation are reported for each condition, the modified index of the CRS-R is reported on the second row for each patient. N/A—Not assessed.

ID	Age	Sex	Diagnosis	TSI	Etiology	Order of Sessions	Measure	Pre Sham	Post Sham	Pre tDCS	Post tDCS	Pre tPCS	Post tPCS
ID01	71	F	MCS+	12.8	Stroke	sham tPCS tDCS	CRS-R	15	13	10	15	14	14
							Modified Index	49.6	47.5	3.5	64.9	55.9	49.3
ID02	31	F	MCS+	5	Anoxic	tDCS tPCS sham	CRS-R	8	10	9	9	9	9
							Modified Index	14.9	31.5	16	15.9	23.2	15.2
ID03	50	F	MCS−	4	Stroke	tPCS tDCS sham	CRS-R	4	2	2	4	7	5
							Modified Index	3.4	1.37	1.37	1.3	21.1	11.8
ID04	63	M	MCS−	2.5	TBI	sham tPCS	CRS-R	5	6	N/A	N/A	3	3
							Modified Index	19.8	6.5	N/A	N/A	3.1	3.1
ID05	47	F	MCS−	3	Stroke	sham tDCS tPCS	CRS-R	7	5	9	6	7	8
							Modified Index	5.8	3.8	15.2	4.8	6.9	7
ID06	65	F	MCS+	5.8	Stroke	tPCS sham tDCS	CRS-R	5	4	5	10	4	7
							Modified Index	4.5	3.5	4.5	29.8	3.4	13.1
ID07	60	F	eMCS	33	TBI	tDCS sham tPCS	CRS-R	14	15	18	19	17	16
							Modified Index	55.8	57.6	82	90	73.6	65.2
ID08	34	M	MCS−	4.5	TBI	tPCS tDCS sham	CRS-R	8	10	11	10	10	10
							Modified Index	22.2	23.6	24.6	23.6	23.6	23.6
ID09	51	F	MCS−	4	Stroke	tDCS sham tPCS	CRS-R	10	10	9	8	9	11
							Modified Index	23.5	23.6	22.5	21.5	23.2	32.6
ID10	23	M	MCS+	2	TBI	sham tPCS tDCS	CRS-R	13	15	17	18	15	15
							Modified Index	4.2	57	66.3	74.6	49.6	49.6
ID11	53	F	MCS−	3	Stroke	sham tDCS tPCS	CRS-R	7	6	7	8	6	5
							Modified Index	14.5	13.5	5.9	7	6.2	4.5
ID12	56	M	eMCS	3	Stroke	tDCS tPCS sham	CRS-R	20	19	17	18	20	20
							Modified Index	91.3	82.3	66.3	74.6	91.3	91.3

**Table 2 brainsci-12-00429-t002:** Details on EEG Connectivity data and behavioral data. Cells report median and (IQR25—IQR75) of post- and pre-differences of EEG data (each band and region for each condition; for example, alpha frontal median for sham condition was −0.08 and IQR25 −0.12, IQR75 0.01) and behavioral data (CRS-R total scores and modified index for each condition. For example, the median of CRS-R total scores for sham condition was −0.5 and IQR25 −1, IQR75 1.25). The last column reports the Chi-squared and *p*-value (between parentheses) for the Kruskal–Wallis comparing that band and region between conditions for the EEG data and the Chi-squared and *p*-value for the Kruskal–Wallis between conditions for behavioral data.

	Sham	tDCS	tPCS	Chi-Squared (*p*-Value)
Alpha frontal	−0.08 (−0.12; 0.01)	0.10 (−0.016; 0.15)	0.02 (−01; 0.09)	4.90 (0.08)
Alpha parietal	0.000 (−0.02; 0.06)	0.045 (0.001; 0.12)	0.011 (−0.04; 0.13)	0.673 (0.71)
Theta frontal	−0.037 (−0.09; 0.09)	0.142 (−0.05; 0.19)	−0.122 (−0.20; 0.01)	5.697 (0.05)
Theta parietal	−0.12 (−0.20; 0.00)	−0.04 (−0.19; 0.03)	−0.01 (−0.09; 0.00)	0.08 (0.64)
CRS-R total score	−0.5 (−1; 1.25)	1 (−0.5; 1.5)	0 (−0.25; 0.25)	1.47 (0.49)
CRS-R modified index	−1.04 (−2.09; 1.47)	4.52 (−0.26; 10.42)	0 (−6.93; 0)	4.11 (0.13)

**Table 3 brainsci-12-00429-t003:** Detailed results from the exploratory analysis testing [tPCS + sham] against tDCS condition. Cells report the median and (IQR25—IQR75) of each band and region for each condition. For example, the alpha frontal median for the [tPCS + sham] condition was −0.002 and IQR25 −0.107, IQR75 0.051. The last column reports the Wilcoxon index and *p*-value (between parentheses) for the Wilcoxon signed-rank statistical tests. tPCS = transcranial pulsed current stimulation, sham = sham, tDCS = transcranial direct current stimulation. The * indicates significance of *p*-value.

	([tPCS + sham])	tDCS	W (*p*-Value)
Alpha frontal	−0.00 (−0.10; 0.05)	0.10 (−0.01; 0.15)	71 (0.11)
Alpha parietal	0.00 (−0.02; 0.11)	0.04 (0.00; 0.12)	96 (0.58)
Theta frontal	−0.060 (−0.16; 0.02)	0.14 (−0.05; 0.20)	59 (0.04) *
Theta parietal	−0.046 (−0.15; 0.00)	−0.03 (−0.19; 0.03)	104 (0.82)

## Data Availability

Not applicable.

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
