# Peer review of "Transcranial Pulsed-Current Stimulation versus Transcranial Direct Current Stimulation in Patients with Disorders of Consciousness: A Pilot, Sham-Controlled Cross-Over Double-Blind Study"

_brainsci, 2022, doi:10.3390/brainsci12040429_

Round 1

Reviewer 1 Report

1) A methodological suggestion for future researches could be  introduced by the Authors in the Discussion (or Conclusions): any DoC patient enrolled in a study for neurostimulation/neuromodulation should previously undergo a detailed evaluation of the residual functional brain connectivity by means of electrophysiological (EEG, EPs, TMS, TMS-EEG) and/or  neuroimaging (fMRI) techniques.

2) pag. 3, line 135: point d is without text.

3) pag. 10, Fig. 4: no comment to this figure is available.

4) pag. 12, line 456: no reference is provided for the mentioned study.

5) the headline Discussion is missing.

6) a number of inaccuracies (single words) are present in the text.

Author Response

Comments and Suggestions for Authors 

1) A methodological suggestion for future researches could be introduced by the Authors in the Discussion (or Conclusions): any DoC patient enrolled in a study for neurostimulation/neuromodulation should previously undergo a detailed evaluation of the residual functional brain connectivity by means of electrophysiological (EEG, EPs, TMS, TMS-EEG) and/or  neuroimaging (fMRI) techniques. 

We thank very much the reviewer for this insightful suggestion. We have added (Line 407, page 10) a sentence on biomarkers screening (with a focus on theta functional connectivity as it is the only previously observed result that could be replicated in our study) for non-invasive brain stimulation enrolment. 

It now reads “Testing for the presence at baseline of theta band connectivity could also be a good predictor of the patient’s likelihood of benefitting from the stimulation. For this reason, it might be worth adding this criterion to inclusion criteria for future studies”. 

  •  pag. 3, line 135: point d is without text. 

We have corrected the typo.  

  • pag. 10, Fig. 4: no comment to this figure is available. 

We have corrected this omission and added the legend for Figure 4. 

  • pag. 12, line 456: no reference is provided for the mentioned study. 

We have added the reference matching the mentioned study on pag. 12, line 441. 

  • the headline Discussion is missing. 

We have added the corresponding headline. 

  • a number of inaccuracies (single words) are present in the text. 

We are very sorry for the remaining inaccuracies and typos. We have carefully proofread the manuscript and corrected all typos.  

Reviewer 2 Report

In a neat pilot paradigm, the authors present first evidence for tPCS being a feasible and safe technique for treatment of patients with DoC. Overall, this is an interesting and a well-designed study. My concerns are mainly methodological. As there seems to be a formatting issue it was not possible to fully evaluate the Results/Discussion sections. Also, I could not access the Supplementary Materials.

Specific comments:

How were the stimulation electrodes attached, did the authors use a saline solution, a gel or a paste?

Line 183: For tPCS the authors specify that the stimulation electrodes were placed below the EEG cap, while in this section they do not specify the montage for the tDCS condition. The montage is specified for both stimulations later on in section “Secondary goal”. For consistency sake, I would recommend to provide this information either for both or for non of the stimulations in this section. On a similar note, I found it a little confusing that the specifications of the EEG recordings are detailed in the section devoted to “Secondary goal”.

Line 206: Do the authors mean 25 recording electrodes?

Line 210: What do the authors mean by two references. Do they mean the reference and the ground or the linked-mastoids/ears reference? – Please specify.

Line 223: The authors state: “Data were filtered from 1Hz to 30Hz”- please specify if a low- and high-cutoff filters or a band-pass filter was/were applied.

Lines: 225-226: The following sentence is confusing: “Visual inspection and independent component analysis (ICA) were used to remove eyeblink and muscle artifacts.” Visual inspection and ICA are two different steps and the performance of an ICA algorithm (depending on the quality of the data) may largely depend upon the visual inspection step.  Please specify what entailed the visual inspection of the data (e.g., identification of bad channels/bad stretches that were excluded for the ICA and the channels were interpolated after the ICA) and indicate which ICA algorithm was applied.

Line 227: What do the authors mean by the “overall average of electrodes”, do they refer to the average reference? Please specify.

Lines: 227-228: The authors state: “If clean epochs extracted were not enough (i.e. <90 epochs of 2 seconds) to perform the statistical analyses, the patient was discarded from the EEG analyses.” – Please specify how many data sets were discarded from further analyses or refer the reader to Fig. 2 where this information can be found.

Lines: 235-236: The authors focus their analyses on frontal and parietal electrodes, which is fine unless a rationale for this choice is clearly outlined.

Lines: 244-245: Similar as above, the authors write “(…) based on literature” but provide no reference.

Lines: 237-238: Do the authors mean all electrode pairs? How did they correct for multiple comparisons?

Line 294: “Fig. 2 Flow chart of patieFigure 2. nts’ enrolment.” – please correct. There seems to be a similar mistake in the heading of Figure 3 (Line 334).

Table 1 – I believe that the readability of the table could be increased by an addition of horizontal lines between the patients.

I am confused about the values reported in Table 2 and the examples provided in the table caption – please carefully revise these.

There seems to be a formatting problem that starts at line 363, i.e.,:

- Figure 4 has no caption and it is not referenced in the running text.

- Part of the Results/Discussion sections seem to be missing, hence cannot be evaluated.

On a similar note, I was not able to evaluate the Supplementary Materials as they could not be downloaded “Error 404 - File not found”.

Thee also seems to be a problem with references in the Discussion section – please carefully revise.

Lines 443-445: While it is true that there is a lack of studies, hence it is difficult to speculate on the effectiveness of tPCS, I believe that the authors could expand a little more on the possible limitations of their study instead of solely listing them. E.g., do the authors believe that the absence of neuronavigation could have influenced the results? If yes, what is the evidence from previous stimulation studies.

Lines 475-480: The authors acknowledge that the pipeline for the extraction of FC values has not been validated with a 27-electrode system. In general, what is the authors take on assessing FC with 27 channels? This is not meant as a critique as the study design is challenging enough and the authors acknowledge the pilot nature of their study, yet the authors could shortly discuss this in light of previous literature.

Please carefully proofread the manuscript for English grammar e.g.:

Lines: 26, 34, and 35 of the abstract

and typos:

Line 422 “ypothesized” instead of hypothesized

Line 431 “id” instead of did

These are only few examples.

Author Response

Comments and Suggestions for Authors 

“In a neat pilot paradigm, the authors present first evidence for tPCS being a feasible and safe technique for treatment of patients with DoC. Overall, this is an interesting and a well-designed study. My concerns are mainly methodological. As there seems to be a formatting issue it was not possible to fully evaluate the Results/Discussion sections. Also, I could not access the Supplementary Materials.” 

We thank the reviewer very much for his/her insightful comment on our study. We regret the formatting issue that he/she has encountered. We think there was a technical issue when uploading the file on the journal website and we hope that this new version will not present any problem and the reviewer will be able to access all parts of the manuscript. We also want to stress that there is no supplementary material file for this study and that that option should not appear on the website anymore.  

  

Specific comments: 

  • How were the stimulation electrodes attached, did the authors use a saline solution, a gel or a paste? 

We have added this information in the method section: page 4, Line 183  

 It now reads “Both tPCS and tDCS electrodes were fixed using a conducting paste (Ten20, Weaver and Company) to reduce impedances and were placed under the EEG cap..” 

  • Line 183: For tPCS the authors specify that the stimulation electrodes were placed below the EEG cap, while in this section they do not specify the montage for the tDCS condition. The montage is specified for both stimulations later on in section “Secondary goal”. For consistency sake, I would recommend to provide this information either for both or for non of the stimulations in this section. On a similar note, I found it a little confusing that the specifications of the EEG recordings are detailed in the section devoted to “Secondary goal”. 

Regarding the “placed under the cap” suggestion we have modified the text to give this information (page 4, line 182) 

Regarding the EEG recordings specifications, we understand the confusion it might entail. However, in this study EEG (together with behavioral assessment) was a secondary outcome and this is the reason why in methods it was placed in that section. 

We have now removed the subheadings “primary goal” and “secondary goal” throughout the whole methods section to avoid the confusion  

  • Line 206: Do the authors mean 25 recording electrodes? 

Indeed, we have corrected with “25 recording electrodes and 2 references” (page 5, line 203-4) 

  • Line 210: What do the authors mean by two references. Do they mean the reference and the ground or the linked-mastoids/ears reference? – Please specify. 

One reference, one ground. We have specified this page 5, line 208 

  

  • Line 223: The authors state: “Data were filtered from 1Hz to 30Hz”- please specify if a low- and high-cutoff filters or a band-pass filter was/were applied. 

Yes, we have applied a low pass and high pass filter at 1 and 30 Hz, we have corrected the sentence in page 5 line 219  

It now reads : “Data were filtered with a high-pass filter and low-pass filter, respectively set at 1Hz and 30Hz.” 

  • Lines: 225-226: The following sentence is confusing: “Visual inspection and independent component analysis (ICA) were used to remove eyeblink and muscle artifacts.” Visual inspection and ICA are two different steps and the performance of an ICA algorithm (depending on the quality of the data) may largely depend upon the visual inspection step.  Please specify what entailed the visual inspection of the data (e.g., identification of bad channels/bad stretches that were excluded for the ICA and the channels were interpolated after the ICA) and indicate which ICA algorithm was applied. 

We have added that visual inspection was used to identify artefactual epochs and noisy channels, which were then removed. In a next step, an Infomax ICA algorithm was used to remove eyeblink and muscle artifacts from the data. 

It now reads “Visual inspection was used to identify and remove artifactual signal epochs and noisy channels. An independent component analysis (ICA) based on the Infomax algorithm41 was used to remove eyeblink and muscle artifacts (based on their temporal, frequency and spatial distribution)” on page 5, line 221-24. 

  • Line 227: What do the authors mean by the “overall average of electrodes”, do they refer to the average reference? Please specify. 

We thank the reviewer for that hint and specified that electrodes were re-referenced to the average taken from all electrodes. This process is known as average re-referencing. 

  • Lines: 227-228: The authors state: “If clean epochs extracted were not enough (i.e. <90 epochs of 2 seconds) to perform the statistical analyses, the patient was discarded from the EEG analyses.” – Please specify how many data sets were discarded from further analyses or refer the reader to Fig. 2 where this information can be found. 

We specified in the text (line 228) that one patient had to be discarded from EEG analyses due to excessive artifacts in the EEG data (less than 90 epochs of good data quality) and referred the reader to Fig. 2. 

It now reads “If clean epochs extracted were not enough (i.e. <90 epochs of 2 seconds) to perform the statistical analyses, the patient was discarded from the EEG analyses. One patient was discarded from EEG analyses for this reason (see Fig. 2 for details). 

  • Lines: 235-236: The authors focus their analyses on frontal and parietal electrodes, which is fine unless a rationale for this choice is clearly outlined  

We have added in line 236, that the rationale for choosing frontal and parietal electrodes follows the mesocircuit model assuming frontal and parietal connectivity on a cortico-cortical level as being relevant for recovery from disorders of consciousness (Edlow, Claassen, Schiff, & Greer, 2021; Schiff, 2010). 

It now reads “We focused the analyses on frontal electrodes (i.e., F3, F4, F7, F8, Fc1, Fc2, Fc5, Fc6, Fp1, Fp2, Fz) and parietal electrodes (i.e., Cp5, Cp6, P3, P4, Pz), as this follows the mesocircuit hypothesis on the recovery from disorders of consciousness42,43.” 

  • Lines: 244-245: Similar as above, the authors write “(…) based on literature” but provide no reference. 

We added a sentence (line 280) referring to the mesocircuit model of consciousness as literature basis for the selection of brain regions for the EEG analyses. 

It now reads in line 245 “We focused the analyses on frontal and parietal regions, using the same electrodes as for power, based on literature regarding the meso-circuit model42,43 

  • Lines: 237-238: Do the authors mean all electrode pairs? How did they correct for multiple comparisons? 

The connectivity values between pairs of electrodes were used to build a n x n connectivity matrix (n being the number of electrodes in the frontal or parietal region). The median of these connectivity values for each frequency band was computed. We specified this procedure in the text on page 5, line 238-40. 

The text now reads “The connectivity between pairs of channels was calculated for each band, in channels encompassing the frontal and parietal region, using the debiased weighted Phase Lag Index (dwPLI)” 

There is no need to correct for multiple comparisons as these values are not compared against each other using statistical tests. This procedure aimed at averaging the connectivity per region (frontal and parietal) and frequency band (alpha and theta). A correction for multiple comparisons was relevant in a later step when testing for effects of condition, region and frequency band, as mentioned on page 6, line 327 “The same analysis was performed for connectivity data. P values were considered significant after correcting for multiple comparisons for the three conditions, three regions and two power bands using a Bonferroni correction (p<0,003).” 

  • Line 294: “Fig. 2 Flow chart of patieFigure 2. nts’ enrolment.” – please correct. There seems to be a similar mistake in the heading of Figure 3 (Line 334). 

We have corrected the mistake of formatting. 

  • Table 1 – I believe that the readability of the table could be increased by an addition of horizontal lines between the patients. 

We thank the reviewer for the suggestion and we have added horizontal lines between patients in Table 1. 

  • I am confused about the values reported in Table 2 and the examples provided in the table caption – please carefully revise these. 

We thank the reviewer for the comment and we have revised the numbers reported, there was indeed an error in the caption of Table 2 that has been corrected. Moreover, to improve readability of the table we have replaced the “-” between the IQR25 and IQR75 with a “;” in order not to confuse it with a minus sign. 

  • There seems to be a formatting problem that starts at line 363, i.e.,: 

Indeed, when adapting the word document to the template of the journal the formatting encountered some issues that weren’t noticed by the authors before submitting. We are sorry for the inconvenience and have revised the format of the entire manuscript. 

  • - Figure 4 has no caption and it is not referenced in the running text. 

We have added the caption that was indeed cut off due to a formatting problem. 

  • Part of the Results/Discussion sections seem to be missing, hence cannot be evaluated. 

On a similar note, I was not able to evaluate the Supplementary Materials as they could not be downloaded “Error 404 - File not found”. 

We regret this inconvenience. When uploading the manuscript on the website, the formatting must have cut some small parts of the text. We have solved the problem in the current version and we hope the reviewer will be able to access all parts of the manuscript now. Note that there is not a supplementary material file for this study. 

  • Thee also seems to be a problem with references in the Discussion section – please carefully revise. 

We have revised the references in the discussion and we have solved the issue. 

  • Lines 443-445: While it is true that there is a lack of studies, hence it is difficult to speculate on the effectiveness of tPCS, I believe that the authors could expand a little more on the possible limitations of their study instead of solely listing them. E.g., do the authors believe that the absence of neuronavigation could have influenced the results? If yes, what is the evidence from previous stimulation studies. 

We thank the reviewer for this precious suggestion. We have expanded the limitations part of the discussion, with a focus on the absence of neuronavigation procedure in the current literature ( page 12, lines 455-61) and added a reference to support the hypothesis. 

It now reads : “Regarding the lack of neuronavigation, it has been suggested26 that montages tailored to the patients’ specific brain lesions might have more probabilities of success, since the structural preservation of brain areas targeted by the stimulation seem to be a crucial factor to respond to stimulation52. In the present study, we could not use this approach as not all patients underwent recent neuroimaging investigation that could have helped us to adapt tCS montages to each patients. However, this should definitely be considered for future studies.” 

  • Lines 475-480: The authors acknowledge that the pipeline for the extraction of FC values has not been validated with a 27-electrode system. In general, what is the authors take on assessing FC with 27 channels? This is not meant as a critique as the study design is challenging enough and the authors acknowledge the pilot nature of their study, yet the authors could shortly discuss this in light of previous literature. 

We argued for the feasibility in the evaluation of FC in this setup (27 electrodes) by citing the original authors, who successfully reduced the electrodes down to 11 without substantially reducing the meaningfulness of the dwPLI connectivity. Another study is now cited successfully applying a familiar measure of connectivity (wPLI) to datasets of eight EEG electrodes in DoC patients. 

On page 13, line 480-84 the text now reads “However, Chennu and colleagues reported that the median dwPLI connectivity was a relatively stable index against reducing the number of electrodes down to at least 11 electrodes. Another study on multichannel tDCS to reduce hypertonia in DoC patients found an effect of the treatment on beta connectivity with only eight EEG recording channels55 

  • Please carefully proofread the manuscript for English grammar e.g.: 

Lines: 26, 34, and 35 of the abstract 

and typos: 

Line 422 “ypothesized” instead of hypothesized 

Line 431 “id” instead of did 

These are only few examples. 

We thank the reviewer for the thorough inspection of the manuscript and we are sorry for the amount of typos still present in the submitted work. Most of them were a result of the formatting issue, all of them (hopefully) have been corrected now.  

Reviewer 3 Report

General comment:

This is a clinically important research. In the current research, the authors investigated the efficacy of tDCS and its variants to manage Disorders of Consciousness (DoC). Generally, the manuscript is well written. I had only a few comments to be addressed.

  1. The authors provided a comprehensive introduction about the DoC. I appreciated with this effort, which would make the ordinary clinicians well recognize the DoC. However, in contrary, the authors only provide few statement about the non-invasive neuromodulation. To date, there have been several important non-invasive neuromodulation developed to manage neuropsychiatric disease. For example, the repetitive transcranial magnetic stimulation (rTMS), theta burst stimulation (TBS, one of the variant of rTMS), non-invasive vagus nerve stimulation, tDCS, and transcranial random noise stimulation (tRNS, one of the variant of tDCS). Further, there have been several important network meta-analyses/meta-analyses/randomized controlled trials (RCTs) addressing the efficacy and safety of those non-invasive neuromodulation in neuropsychiatric disease, such as dementia/minimal cognitive impairment (PMID: 30229671),nicotine cognition in brain disorder (PMID: 33070785), obesity, Alzheimer's disease (PMID: 33115936), tinnitus and treatment resistant depression (PMID: 31863873), minimal cognitive impairment + Parkinson (PMID: 33408684). Therefore, I would strongly recommend the authors to cite all these references and make a brief statement about the WIDE application of non-invasive neuromodulation in neuropsychiatric disease.
  2. In the exclusion criteria, I could recognize the necessity of excluding seizure. However, why the authors excluded “untreated seizure patients” but not “all seizure patients”? In addition, although the most tDCS experts could realize the reason of those exclusion criteria, some ordinary readers might not be familiar with the contraindication of tDCS/tPCS. I would recommend the authors make a brief explanation of their exclusion criteria.
  3. Since this is a crossover trial, the adequate washout period is important. The authors said “Each session was separated by a minimum of 5 days to avoid any carryover effect”. It is good. However, I would recommend the authors to make a brief description how they determine a “minimum of 5 days” to be adequately wash-out. In addition, I noticed that all the patients only receive ONE session of stimulation of each treatment arm. I would recommend the authors to explain why they choose ONE session but not a whole course.
  4. In the non-invasive neuromodulation study, an adequate blindness is important. The most RCTs of non-invasive neuromodulation study would test the blindness of study design at the end of trial (i.e. to ask patients which treatment arms did they feel to be assigned). Did the authors do this test?
  5. Although there is statistically insignificant result, we would observe a potential beneficial effect. I would agree that small sample size might be one of the reason. Another potential issue about this result might be derived from the neural noise hypothesis. Please check the reference PMID: 21685932 and make a brief discussion about this.
  6. Finally, as addressed in the previous comment, the insufficient sessions of stimulation might be another reason of insignificantly different results. Please address it.

Author Response

Comments and Suggestions for Authors 

General comment: 

This is a clinically important research. In the current research, the authors investigated the efficacy of tDCS and its variants to manage Disorders of Consciousness (DoC). Generally, the manuscript is well written. I had only a few comments to be addressed. 

We thank the reviewer for the kind words on our work and we appreciate his/her feedback on our manuscript. 

  

  • The authors provided a comprehensive introduction about the DoC. I appreciated with this effort, which would make the ordinary clinicians well recognize the DoC. However, in contrary, the authors only provide few statement about the non-invasive neuromodulation. To date, there have been several important non-invasive neuromodulation developed to manage neuropsychiatric disease. For example, the repetitive transcranial magnetic stimulation (rTMS), theta burst stimulation (TBS, one of the variant of rTMS), non-invasive vagus nerve stimulation, tDCS, and transcranial random noise stimulation (tRNS, one of the variant of tDCS). Further, there have been several important network meta-analyses/meta-analyses/randomized controlled trials (RCTs) addressing the efficacy and safety of those non-invasive neuromodulation in neuropsychiatric disease, such as dementia/minimal cognitive impairment (PMID: 30229671),nicotine cognition in brain disorder (PMID: 33070785), obesity, Alzheimer's disease (PMID: 33115936), tinnitus and treatment resistant depression (PMID: 31863873), minimal cognitive impairment + Parkinson (PMID: 33408684). Therefore, I would strongly recommend the authors to cite all these references and make a brief statement about the WIDE application of non-invasive neuromodulation in neuropsychiatric disease. 

We appreciate very much the reviewer’s suggestion of adding more information on the wide use of NIBS in neuroscience and neuropsychology. Indeed, we had not gone into too much detail about it. We have provided a brief overview of NIBS state of the art use in some conditions in page 2 line 58-64. 

It now reads: “Neuromodulation is a broad term that refers to different brain stimulation techniques that can be either invasive or non-invasive. It is now widely used to treat several neuropsychological conditions as an alternative to or for people that are resistant to pharmacological treatments. Some stimulation approaches have proved to be such a valid option that are currently also FDA approved in several countries.”  

  • In the exclusion criteria, I could recognize the necessity of excluding seizure. However, why the authors excluded “untreated seizure patients” but not “all seizure patients”? In addition, although the most tDCS experts could realize the reason of those exclusion criteria, some ordinary readers might not be familiar with the contraindication of tDCS/tPCS. I would recommend the authors make a brief explanation of their exclusion criteria. 

We understand the rationale behind the reviewer’s question. Many patients with DoC are treated for seizures and excluding all patients with a history of seizure would have had a huge impact on enrollment. Previous studies performing tES in patients with DoC have included also patients treated for seizures and have not found any side effect. For this reason we believed safe to include patients treated for seizures but not those that presented seizures and were not treated for them.  

We have added a sentence in the inclusion criteria explaining this. 

Page 3, Line 136-8 now reads: “ Note that the reason why only excluded patients presenting untreated seizures is that previous tES trials on patients with DoC have enrolled patients treated for seizures and have not found any side effect.” 

  • Since this is a crossover trial, the adequate washout period is important. The authors said “Each session was separated by a minimum of 5 days to avoid any carryover effect”. It is good. However, I would recommend the authors to make a brief description how they determine a “minimum of 5 days” to be adequately wash-out. In addition, I noticed that all the patients only receive ONE session of stimulation of each treatment arm. I would recommend the authors to explain why they choose ONE session but not a whole course. 

We thank the reviewer for this detailed comment and suggestion. The washout period of 5 days has been based on literature (Nitsche et al 2001) and on similar studies. TDCS effects should not last more than 2 hours after the end of stimulation, as we are not aware if this is true also for tPCS yet, we have decided to increase this to 5 days as a precaution. We have added this explanation at page 4 lines 155-7  

Regarding the choice of performing a single session compared to repeated sessions, repeated sessions studies pose many more problems on enrollment and dropout rates, apart from taking a lot longer to be completed. We are aware of the higher rates of responders for repeated sessions, however since this pilot study was designed to assess feasibiliy and safety of tPCS mostly we decided to start with a single session of each condition. We do agree that future studies, once established the feasibility of the stimulation) should focus on repeated sessions rather than single. We have added a brief explanation on this point in the limitations section of discussion (page 13, lines 485-490) 

  • In the non-invasive neuromodulation study, an adequate blindness is important. The most RCTs of non-invasive neuromodulation study would test the blindness of study design at the end of trial (i.e. to ask patients which treatment arms did they feel to be assigned). Did the authors do this test? 

We could not perform this test for blindness as our patients do not have the ability to communicate. However, the NeuroConn devices report being tested successfully for blindness by healthy subjects and the researcher’s blindness was ensured by the study design (I.e. randomization code and envelopes created by another researcher and different researcher performing the behavioral assessment) 

  • Although there is statistically insignificant result, we would observe a potential beneficial effect. I would agree that small sample size might be one of the reason. Another potential issue about this result might be derived from the neural noise hypothesis. Please check the reference PMID: 21685932 and make a brief discussion about this. 

We thank the reviewer for this insight. We have added a paragraph in the discussion section (page 12, lines 464-72)   

It now reads: “In our case for instance, although the frequency that was chosen for tPCS was based on modelling studies, it might have been not optimal for DoC patients. Based on the neural noise hypothesis, spontaneous frequencies need to be present in the brain at baseline in order to be entrained by neuromodulation53. A frequency range between 6 and 10 Hz (as the one used in our study) was able to produce EEG changes in healthy participants54. We can, thus, hypothesize that stimulating within this range in patients with DoC, that demonstrate predominant activities within the delta-theta range, failed to produce neurophysiological results. In this context, targeting lower frequencies (e.g., 2-6 Hz) could be more effective in DoC..” 

  • Finally, as addressed in the previous comment, the insufficient sessions of stimulation might be another reason of insignificantly different results. Please address it. 

We have addressed this in the limitation part of the discussion (page 13, lines 485-490). 

Round 2

Reviewer 3 Report

The authors had addressed the most comments I mentioned. I think the current version is good to be accepted.